# Robotic Complex for Harvesting Apple Crops

**Oleg Krakhmalev** [1], **Sergey Gataullin** [2], **Eldar Boltachev** [1,*], **Sergey Korchagin** [1], **Ivan Blagoveshchensky** [3] and **Kang Liang** [4]

1 Department of Data Analysis and Machine Learning, Financial University under the Government of the Russian Federation, 4-th Veshnyakovsky Passage, 4, 109456 Moscow, Russia; olegkr64@mail.ru (O.K.); sakorchagin@fa.ru (S.K.)
2 Information Security Department, Financial University under the Government of the Russian Federation, 4-th Veshnyakovsky Passage, 4, 109456 Moscow, Russia; stgataullin@fa.ru
3 Federal State Budgetary Educational Institution of Higher Education, Moscow State University of Food Production, Volokolamsk Highway, Building 11, 125080 Moscow, Russia; igblagov@mgupp.ru
4 Engineering Training Center, Shanghai Polytechnic University, Shanghai 201209, China; kangliang@sspu.edu.cn
* Correspondence: eleldar@mail.ru

**Abstract:** The article deals with the concept of building an automated system for the harvesting of apple crops. This system is a robotic complex mounted on a tractor cart, including an industrial robot and a packaging system with a container for fruit collection. The robot is equipped with a vacuum gripper and a vision system. A generator for power supply, a vacuum pump for the gripper and an equipment control system are also installed on the cart. The developed automated system will have a high degree of reliability that meets the requirements of operation in the field.

**Keywords:** columnar apple trees; tree fruit; packaging automated system; industrial robot



## 1. Introduction

According to the available data, the area occupied by apple crops in Russia has increased by more than 5 times in the last three years. Therefore, the use of robotic complexes for apple crop harvesting has a great prospect.

Modern agriculture has great difficulty in attracting labor during the harvest season. One of the complex challenges is the fruit harvest. Bu et al. [1] found that to solve the difficulties faced by apple harvesters, the development of mechanized and automated apple harvesting and modern orchard structure applications must be accelerated.

At present, automation and new technical means, including industrial robots, are widely implemented in agriculture. Such operations as harvesting, transportation and processing of agricultural products are being automated. Each of these technological operations has its specifics. Different processes have different characteristics and require different automation approaches. Many operations already have examples of automation or are performed through mechanization. For some operations, automation is just beginning to be used [2].

However, results have already been obtained that can provide a theoretical basis for future studies on robotic apple harvesting [3]. To solve this problem, the scientific literature distinguishes three main scientific directions. These include robot programming, computer vision and machine learning.

Tahriri et al. [4] note that robot programming is a complex task whereby the user needs to teach and control the robot to perform the desired action. To address the above problem, an integrated three-dimensional (3D) simulation software and virtual reality (VR) system are developed to simplify and speed up tasks and therefore enhance the quality of manufacturing processes. Liu et al. [5] developed a virtual grip-and-cut model of cluster picking, and corresponding cluster vibration and fruit falling simulations were performed

for robotic harvesting. The development of a toolbox for kinematic and dynamic modeling and analysis of the high degree of freedom is presented in [6].

Zhang et al. [7] presented a five-dimensional fusion model of a digital twin virtual entity for robotics-based smart manufacturing systems to support automatic reconfiguration. Kang et al. [8] presented a robotic vision system to perform fruit recognition and environment modeling for autonomous apple harvesting. Several studies devoted to research on a modifiable development platform for robotic fruit harvesting are presented; this can be used to test specific design choices on different fruit and growing conditions [9,10].

Gongal et al. [11] developed a new sensor system with an over-the-row platform integrated with a tunnel structure that acquired images from opposite sides of apple trees. In [12] Gongal et al. carried out the development of a machine vision system consisting of a color CCD camera and a time-of-flight (TOF) light-based 3D camera for estimating apple size in tree canopies. Mola et al. [13] examined the usefulness of combining RGB-D and radiometric information for fruit detection.

In [14] Mola et al. presented a new technique that uses a mobile terrestrial laser scanner (MTLS) to detect and localize Fuji apples. Sabzi et al. [15] developed a new computer vision algorithm to detect the existing fruits in aerial images of an apple cultivar and estimate their ripeness stage. Fan et al. [16] proposed a novel, multifeature, patch-based apple image-segmentation technique using the gray-centered red–green–blue (RGB) color space. The proposed method can perform with high efficiency and accuracy to guide robotic harvesting.

Saedi et al. [17] developed a convolutional neural network (CNN) and optimized it for fruit recognition based on RGB images. Gao et al. [18] proposed a multiclass apple detection method in dense-foliage fruiting-wall trees based on the faster region-convolutional neural network. Apolo et al. [19] present a rapid-sensing and yield-estimation scheme using off-the-shelf aerial imagery and deep learning. A region-convolutional neural network was trained to detect and count the number of apple fruits on individual trees located on the orthomosaic built from images taken by the unmanned aerial vehicle (UAV). Darwin et al. [20] elucidated the diverse automation approaches for crop yield-detection techniques with virtual analysis and classifier approaches. Zhang et al.'s [21] convolutional neural networks (CNNs) were utilized to identify the tree trunks and branches for supporting the automated excitation location determination.

Zhang et al. [22] proposed three types of adaptive robust synchronous controllers to solve the trajectory tracking problem for a redundantly actuated parallel manipulator. Yu et al. [23] presented an autonomous humanoid robot, namely LABOR, for apple harvesting. Comprised of a binocular camera, a humanoid dual-arm operating system and a mobile vehicle platform, the devised robot is capable of identifying, positioning, grasping and picking up apples via its vision identification and dual-arm harvesting systems.

Successful algorithms to robotize the harvesting process are given in papers [24,25].

Fruit-harvesting automation is difficult due to the sensitivity of fruits to mechanical damage. Mechanical damage to the fruit shortens the shelf life, which makes it impossible to use many automation methods. The Financial University under the Government of the Russian Federation (Moscow) with other agrarian and technical universities conducted research on the automation of fruit and vegetable harvesting [26–29]. The use of a manipulation robot equipped with a vision system and a special vacuum gripper allows for the automation of fruit collection, preserving their integrity.

Recently, colony-shaped apple trees have been planted in most new areas. A columnar apple tree is a natural clone of an apple tree that has no side branches (Figure 1). In British Columbia, in the village of Kelowna, Canada, an old Macintosh apple tree about 50 years old had an unusual branch on it. It had unusually many leaves and fruits and no side branches. This happened in 1964. This spontaneous mutation was not ignored by breeders and was multiplied. Over time, with its help, specialists created columnar apple trees, and both English Kent County plant breeders and specialists from other countries were

working on this plant. In 1976 they managed to obtain the first samples of this kind of apple tree [19].

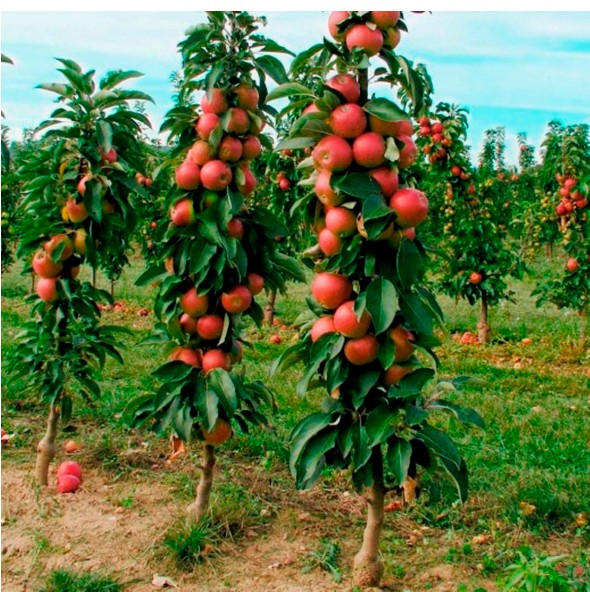

**Figure 1.** Columnar apple trees.

Dwarf varieties, which include columnar apple trees, are less prone to branching than medium-growing (1.5–3 times) and high-growing (3–4 times) varieties. After the tree is 3–4 years old, its side branches stop growing. Such an apple tree will blossom and bear fruit in the second or third year of life. The harvest in the first 5–6 years every year becomes more abundant, but from the 7th–8th year of life of the plant, it is consistently high. Mature trees usually do not grow higher than 2.5–3 m. From each tree 5 to 15 kg of apples can be collected; the weight of the fruit ranges from 150 to 250 g. This type of apple tree is well-suited for automation of the harvest, as the apple trees are not too tall and the fruit along the trunk is accessible to the robot arm, and there is no interference in the form of branches [11,15].

One of the important issues in the automation of fruit harvesting is to ensure the autonomy of the technical system. This is because this system will work in the orchard and will not have access to stationary sources of energy [1,9,23].

The purpose of this project was to develop a robotic complex for apple crop harvesting, composed of elements presented in the market of off-the-shelf industrial automation and robotics products. The use of reliable and proven solutions is a feature of this project, distinguishing it from the majority of experimental works presented in the review.

## 2. Concept

The robotic complex being developed is an automated system for harvesting apple crops, mounted on a mobile platform on wheels (Figures 2 and 3). The complex equipment installed on the mobile platform consists of an industrial robot, a packaging system, vision system, control system, vacuum pump, electric generator and a container. The platform moves in a coupling with a tractor and moves through the garden between the trees. The problem of autonomous power supply of electrical equipment is solved by connecting the electric generator to the power take-off shaft of the tractor.

The basis of the automated system is an industrial robot [6]. The task of the robot is to bring the gripper to the fruit and tear the apple from the tree. This system uses a KUKA KR 30-3 industrial robot (Figure 4). The lifting capacity of the robot is 30 kg. Although this lifting capacity is excessive, it allows apple picking with the vision system, a gripper and a flexible packaging-system element mounted on the robot arm.

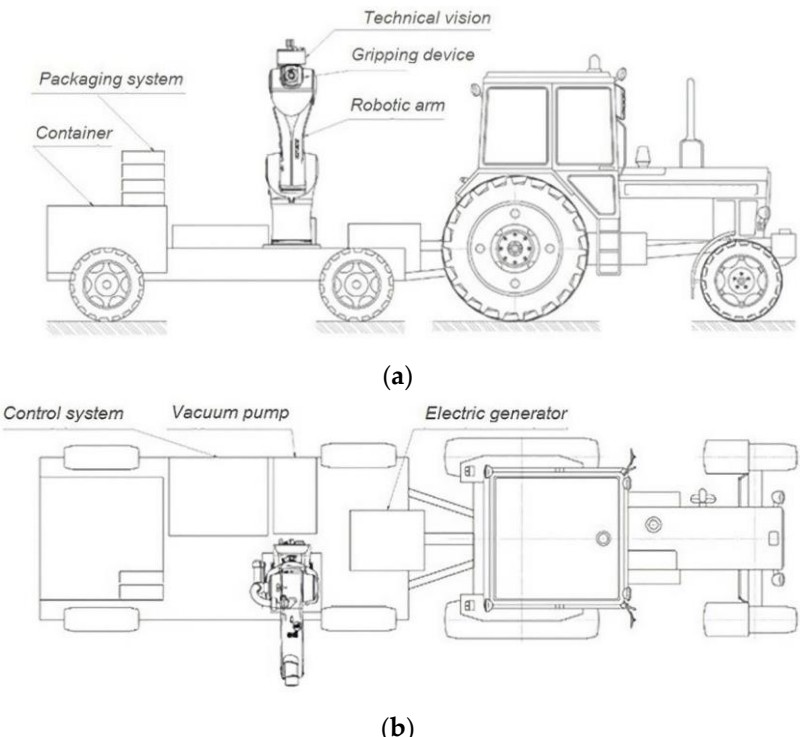

(**a**)

(**b**)

**Figure 2.** Automated fruit-collection system: (**a**) side view; (**b**) top view.

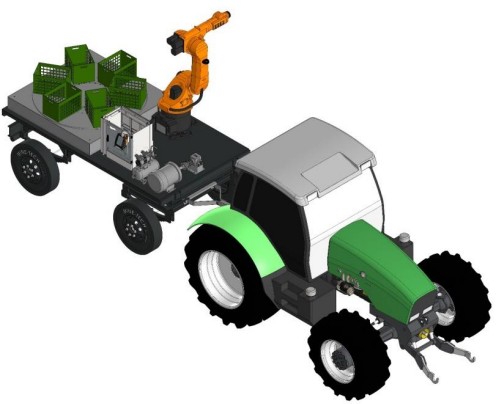

**Figure 3.** Graphic model automated fruit-collection system.

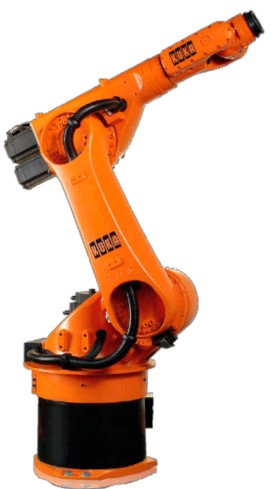

**Figure 4.** Industrial robot KUKA KR 30-3.

An important characteristic when choosing an industrial robot is its workspace (Figure 5); it must be able to "reach" all the apples on the tree, both those that are hanging at the bottom and those that are at the top of the tree.

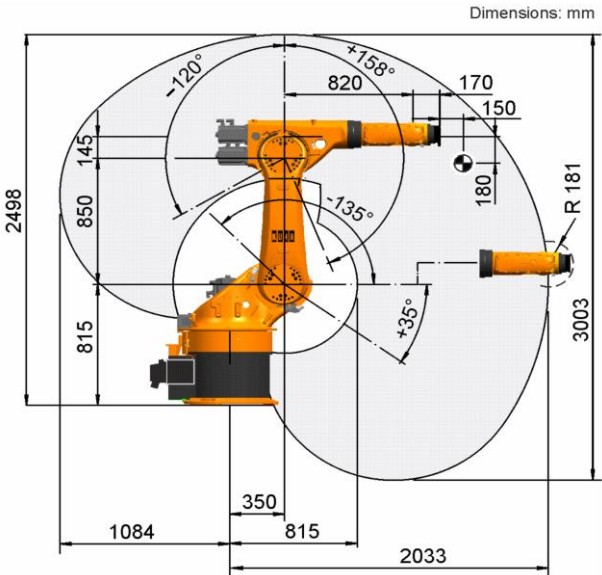

**Figure 5.** Workspace of the robot KUKA KR 30-3.

A vacuum suction cup is used as a gripping device. This type of gripper is well-suited for working with apples because the apple has a smooth, airtight surface. The soft surface of the vacuum suction pad avoids damaging the apple, which is very important for increasing the shelf life of such products. The vacuum grippers are very fast, which means short cycle times and high energy efficiency. This is important in the field. A vacuum generator system is used to supply the gripper robot with the vacuum.

For transporting each removed apple and for intermediate storage of apples, a packaging system is used. The packaging system is a funnel that extends when an apple is picked and torn off to catch the fruit (Figure 6). The packaging system has a flexible sleeve that is used to transport the fruit into the container and also for intermediate storage of apples inside the sleeve when picking low-hanging fruit (Figure 7). For the storage of apples, a container with the dimensions 1120 × 1120 × 770 is used. After filling with fruits, the container can be removed and replaced with a new one. Afterwards, the filled containers are transported to storage or for processing.

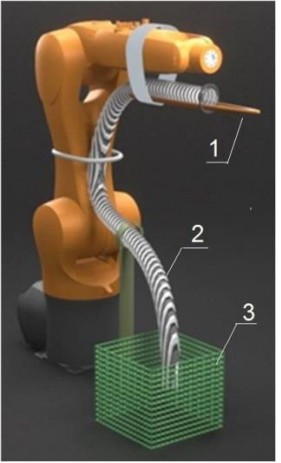

**Figure 6.** Graphic model of the filling system: 1—catching funnel; 2—flexible sleeve; 3—container.

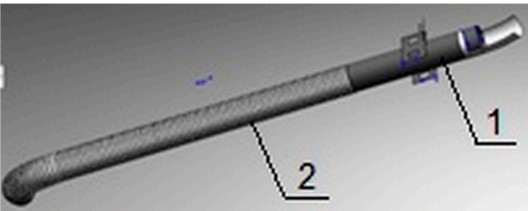

**Figure 7.** Flexible bagging filling system: 1—catching funnel; 2—flexible sleeve.

The robot receives information about the position of apples and their coordinates from the vision system. While picking apples an important task is to determine the position of the fruit, the control system needs to know the coordinates of the apple before it can start the apple picking cycle. This task is solved by the vision system, which is able to recognize apples among many heterogeneous objects, such as branches and foliage, and to determine the coordinates of a particular apple before starting the picking cycle.

Applied scientific results published by the authors earlier will be taken into account in subsequent developments of automated systems [30–37]. It is planned to develop its vision system based on image analysis using an artificial neural network and selection of optimal trajectories of robot gripper movement based on the application of the genetic algorithm. Practical implementation of this project will allow, based on reliable engineering solutions, to reduce the amount of manual labor in agriculture, in particular in the field of growing and processing apple crops.

### 3. Methods

Since fruit picking is a cyclic process, its productivity is determined by the duration of individual operations of the technological cycle. In this case, time-saving in the automation process can be achieved by reducing the time of individual operations and by the simultaneous performance of some operations [3,4].

The technological cycle of fruit gathering includes a sequence of the following operations: 1—location; 2—arm extension; 3—gripping and separating the fruit; 4—returning the arm to its original position; 5—packaging (Figure 8).

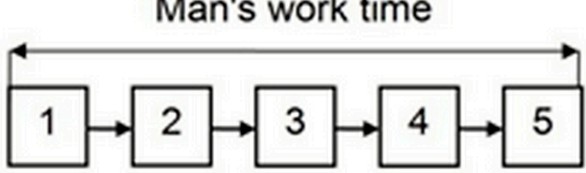

**Figure 8.** Work cycle time (man).

If it is necessary to fully simulate the work of a person, the location must precede the removal of each fruit. A preliminary scanning of the crown volume and mapping of positions of all fruits in it allows to speed up the process considerably. Then, during the gathering, it is necessary to correct the coordinates of each next fruit promptly due to changes in its position, because of displacement of branches due to the removal of a part of fruits from them. In this case, the map correction can be carried out independently of the robot arm action, and therefore the time spent on location is significantly reduced (Figure 9).

The execution time of the second operation of the technological cycle corresponding to the extension of the robot arm equipped with a gripper device depends on the distance between the location point and the fruit selected for removal. It is necessary to take into account the service factor value in the given area of the robot's working space, as well as the maximum accelerations allowed under the working conditions.

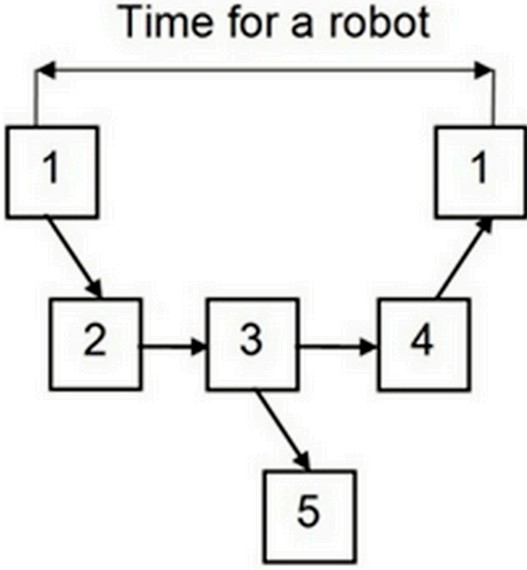

**Figure 9.** Working cycle time (robot).

The reorientation of gardeners to columnar and dwarf orchards greatly facilitates the task of automating the harvesting of apple crops much easier. The fruits on such plants are compactly placed, which allows for the minimizing of distances when moving the gripper. This is also facilitated by the fact that already during the withdrawal of the gripper, the knowledge of the coordinates of the next object to be picked allows the movement along a complex trajectory that provides the fastest exit to the optimal initial position for preparation for the next cycle of work operations.

As for the actual extension of the gripper from the location point to the fetus for gripping, as a rule, it is carried out along a rectilinear trajectory with the maximum possible acceleration. The direction of the approach to the fetus at a certain angle to the horizon (from bottom to top) is ensured.

Capturing and detaching the fruit can be performed in different ways. For example, removal of the gripped fruit can be carried out by tearing off the pre-twisted stalk of the fruit. For this purpose, a four-fingered gripper, which imitates a human hand, grasps the fruit on all sides and tears it off by rotating around its axis [5,10]. However, the best results in terms of time are shown by the vacuum device, which sucks the fetus into the flexible sleeve almost instantly. This completely eliminates the step of releasing the gripper from the fetus, which is immediately transported through the flexible sleeve.

In our research, we decided to combine the advantages of both methods described above and propose a simple gripping device consisting of one vacuum suction cup and a flexible sleeve attached to the robot arm. The vacuum suction cup sticks to the surface of the fetus without damaging it and by combining the pulling force with the rotation, it performs tearing of the fetus. The plucked fetus is lowered in a short movement into the socket of the attached elastic tube, after which the robot arm immediately moves to the starting position for the next fetus removal operation. The cycle shown in Figure 9 is repeated.

The force of detachment of the stalk from the branch is about 2 kgf. This force is greater than the holding force of the apple; therefore we focus on this figure. Taking into account the safety factor of 1.5, the vacuum gripper has to provide a force of not less than 3.0 kgf. The Festo suction pad ESS-30-GT-G1/8 (Figure 10a) is a round, polyurethane elastomeric grip with a diameter of 30 mm and a G1/8 female thread for lifting and transporting workpieces with smooth, airtight surfaces using manipulators. The vacuum generator (ejector) used is the Festo VAD-1/8 (Figure 10b), which has a single-stage design and operates according to the Venturi principle, with a maximum switching frequency of approximately 10 Hz.

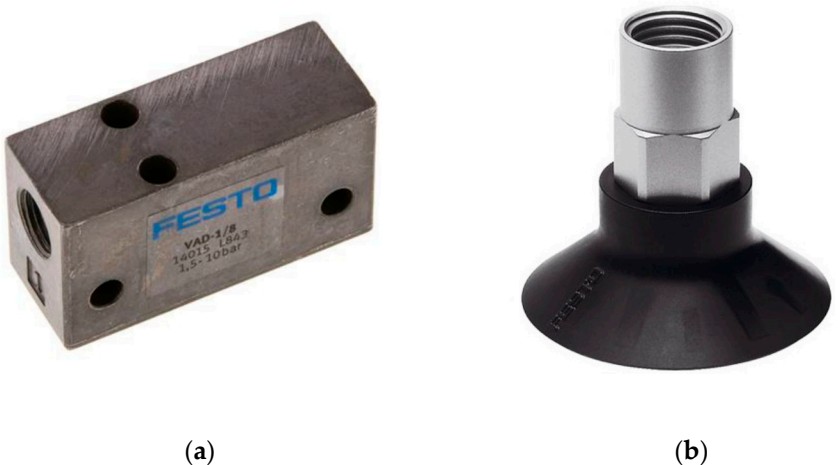

(**a**)　　　　　　　　　　　　　　　　　　　　(**b**)

**Figure 10.** (**a**) Vacuum gripper (suction cup) Festo ESS-30-GT-G1/8; (**b**) vacuum generator Festo VAD-1/8.

The packaging system is a system of transporting apples through a flexible sleeve attached to the robot arm into a container (Figures 6 and 7). This system also controls the level of filling of containers and signals their filling. An ultrasonic level sensor assesses how full the container is.

The vision system solves the following tasks [8,12–14,16–18,20,21,38]:

–　searching for the next fetus in the robot's workspace by changing the orientation of the recognition device;

–　measuring the distance to the observation objects;

–　automatic adjustment of the video sensor depending on the illumination of the working area;

–　comparison of objects with the "model" in memory according to the specified criteria.

The vision system of the developed automated system is based on the software and hardware solutions of Roboception GmbH (Germany), which is a pioneer in the field of 3D-sensor technology [39]. The company's vision system RC Visard 160 recognizes an unstructured environment in real-time and positions objects accurately enough in space (Figure 11). As a result, the robot can fix its position in its workspace with millimeter accuracy and solve tasks quickly and efficiently.

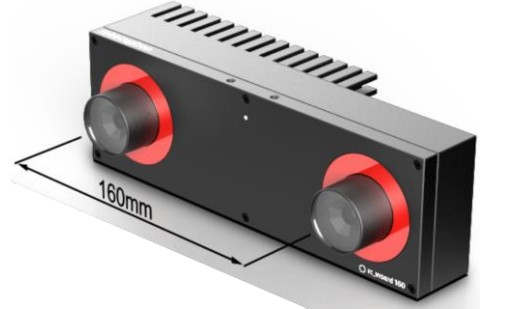
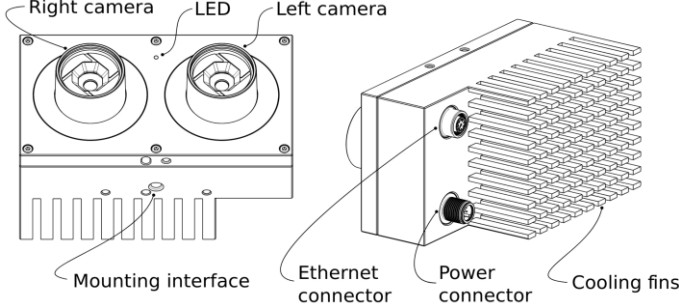

**Figure 11.** Stereo sensor RC Visard 160.

Thanks to integrated RC Visard technology, the resulting images are processed directly in the sensor. The 3D sensor can recognize its surroundings, both in natural light and in low light conditions. Even with fast movements, high accuracy is ensured. The integrated graphics card allows images to be processed directly in the sensor without the need for external calculations. The high-performance hardware and software are designed specifically for use in a robotic environment. The sensor is designed for operation at

temperatures from 0 to 50 °C and complies with IP 54; it also qualifies as Industrial Class A according to EN55011.

When installed on the robot arm, the sensor must be secured with three M4 mounting screws (standard metric) and tightened to 2.5 Nm using a medium-strength thread adhesive, such as Loctite 243. The minimum tread depth is 6 mm.

The RC Visard operation is based on the use of stereo vision realized using the SGM (Semi-Global Matching) method. In 3D stereo vision, information about the scene can be extracted by comparing two images taken from different angles. Different points of the object are in different positions in the images from the two cameras, depending on their distance from the cameras. Very distant object points are at approximately the same position in both images, while very close object points are at different positions in the left and right camera images. The displacement of the object points in the two images is called a mismatch. The greater the mismatch, the closer the object is to the camera. This principle is illustrated in Figure 12.

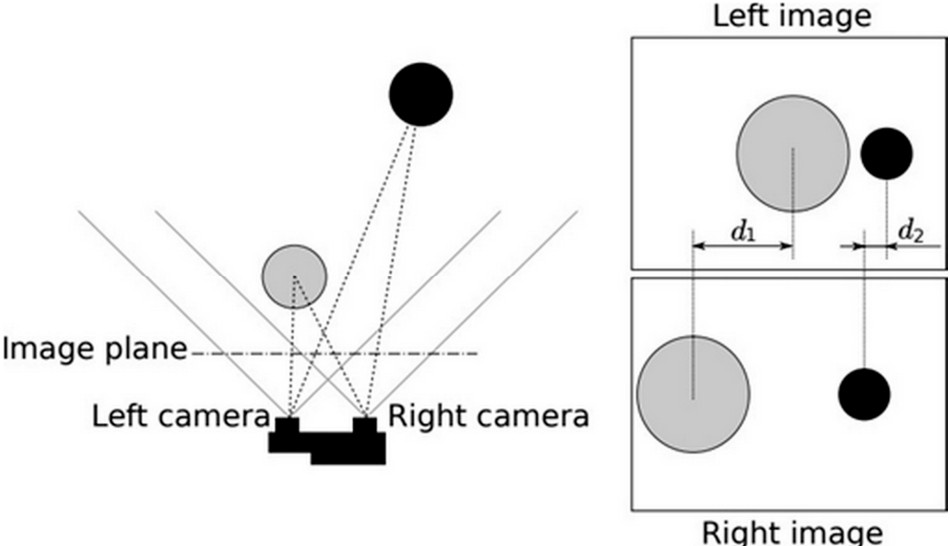

**Figure 12.** Scheme of the stereo vision principle: the more distant object (black) shows less mismatch $d_2$, than nearest object (gray), $d_1$.

The ItemPick software allows you to use the sensor in pick-and-place robotics applications. This software is designed to recognize flat images of objects to pick them up with a vacuum picker. The ItemPick software includes:

– a dedicated page in the RC Visard web interface for easy setup and testing;
– definition of capture areas to select the appropriate planes in the scene;
– a work-area detection function for debris-collection applications to ensure that items are captured only within that area;
– definition of compartments within container cans to place items in a specific order only;
– definition of the surface quality for each captured object;
– sorting objects according to their location, so that the items at the top of the pile are grabbed first.

ItemPick defines the position of the TCP (tool center point) of the vacuum gripper. The orientation of the gripper is the right coordinate system and is defined so that its z-axis is normal to the surface and points inside the object, and the x-axis points along the maximum size of the surface (Figure 13).

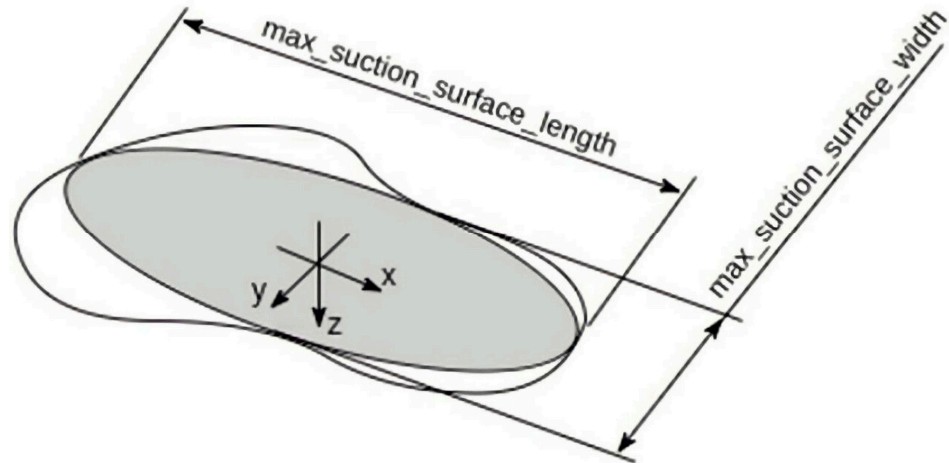

**Figure 13.** Selecting the TCP and orientation of the gripper coordinate system.

The vision system RC Visard is connected to the control system of the industrial robot with an Ethernet cable. The ItemPick app can be launched directly from the SmartPAD control panel of a KUKA robot. KUKA System Software 8.5 and KUKA PerceptionTech must be installed on the robot control panel. To work with RC Visard you will need to:

- perform calibration;
- set up the object recognition;
- define the area in which the robot should work;
- perform a recognition test and optimize the parameters;
- adjust the enclosed sample programs.

A calibration grid is used when performing calibrations. When using a sensor mounted on the robot, it must be securely attached to the flange. The TCP weight must be updated in "TCP Configuration" before starting the calibration, and the calibration grid must be positioned so that it does not move during the calibration (Figure 14).

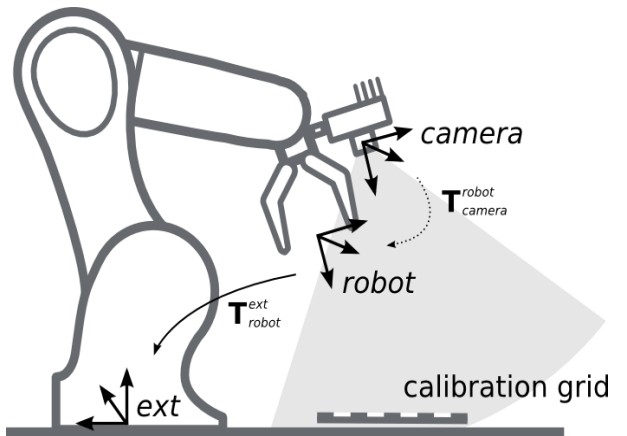
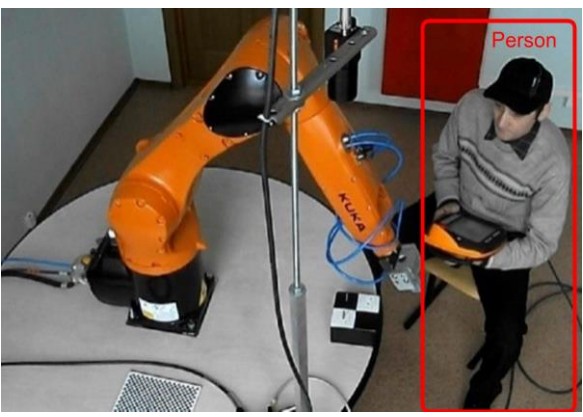

**Figure 14.** Calibration of the vision system: scheme (**left**), process (**right**).

The object recognition is set up by measuring the depth of the image. This places the object in front of the RC Visard and activates the "Depth" tab in the ItemPick web graphical interface. The minimum object size required for the sensor is set at a depth of 300 pixels, which roughly corresponds to an object of 4 × 4 cm at a distance of 1.2 m from the sensor (Figure 15).

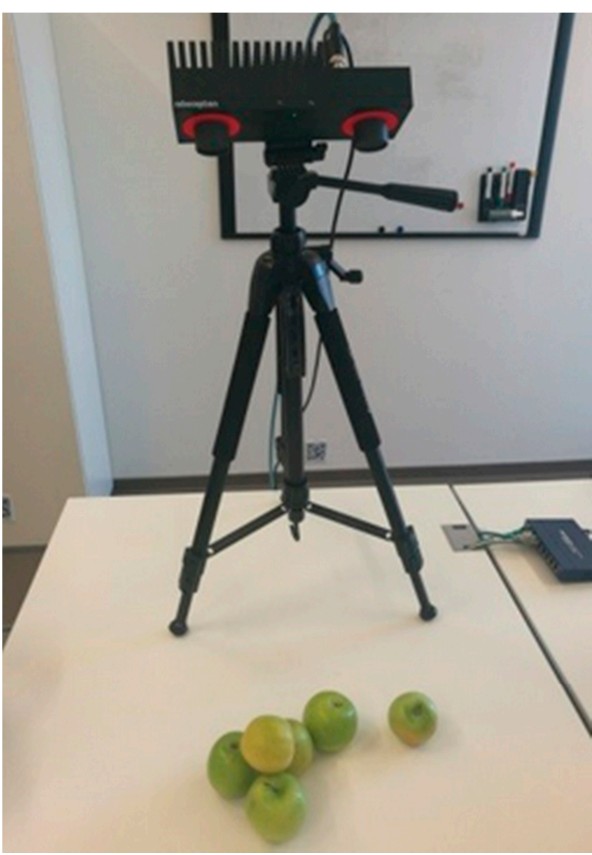

**Figure 15.** Testing the sensor RC Visard 160.

One or more objects are placed in the field of view of the camera (Figure 16). They must meet the following requirements: the surface, shape and weight of the workpiece are suitable for gripping with a vacuum suction cup.

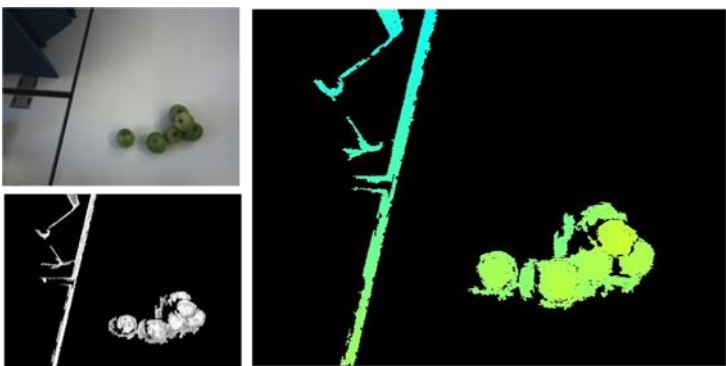

**Figure 16.** Examples of images from the setup shown above: image (**top left**), true image (**bottom left**) and depth image (**right**).

ItemPick provides the best results when the depth image quality is set to high or full. The static mode can be useful in static scenes, but it increases acquisition time (Figure 16). The vacuum suction cup gripping points are calculated using the compute grasps service of the ItemPick module (Figure 17). By default, the compute grasps service returns a maximum of five grasps. The gravity vector is taken into account when sorting the grasps, so the objects at the top of the pile are grasped first.

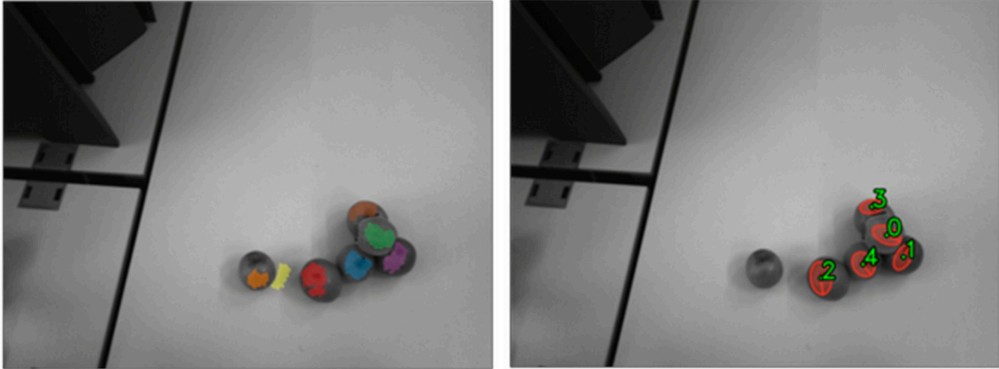

**Figure 17.** The image of the objects on the **left** and the calculated capture points on the **right**.

Figure 18 shows the main elements or the robot complex, which includes the SmartPAD used to program robot movement.

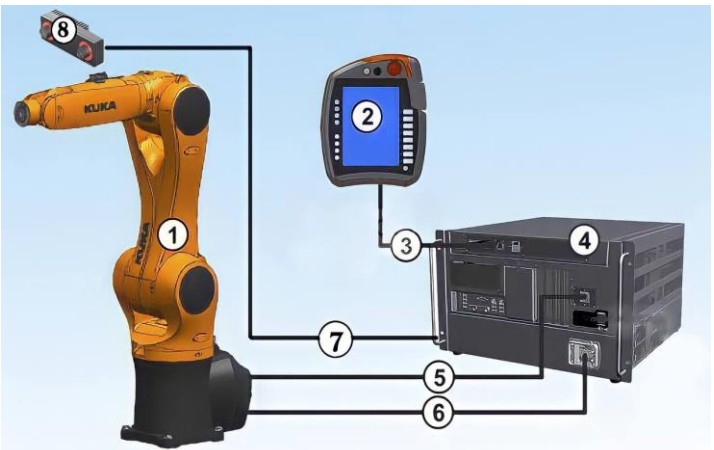

**Figure 18.** 1—Robotic arm (robotic arm), 2—SmartPAD programming console, 3—connection cable, 4—KUKA KR C4 controller, 5—data cable, 6—motor control cable, 7—Ethernet cable, 8—3D sensor RC Visard 160.

## 4. Conclusions and Discussion

The paper presents a project to develop a robotic complex for apple crop harvesting. At first, the concept of such a robotic complex was developed based on the idea of using ready-made technical solutions that have proven themselves in industrial operation conditions.

Based on the developed concept, it was decided to use a KUKA KR 30-3 industrial robot with a payload capacity of 30 kg and a working height of about 3 m. The robot works under the control of the KUKA KR C4 controller. The movements of the robot are programmed via the SmartPAD remote control.

A simple gripper consisting of one Festo ESS-30-GT-G1/8 circular vacuum suction pad made of polyurethane elastomer with a diameter of 30 mm and a G1/8 connection internal thread and a Festo VAD-1/8 vacuum generator with a single-stage design are suggested. This gripper is designed for lifting and carrying workpieces with smooth, airtight surfaces.

The technical vision of the automated system is based on software and hardware solutions of Roboception GmbH, a pioneer in 3D sensor technology. The RC Visard 160 vision system was used, operating in real-time and positioning objects in space with millimeter accuracy.

For transportation and intermediate storage of apples, a bagging system has been developed equipped with a funnel that extends when gripping an apple and tearing it off to catch the fruit. The bagging system has a flexible sleeve that is used to transport the fruit into the container, as well as for intermediate storage of apples inside the sleeve

when picking low-hanging fruit. For storage of apples, a container with dimensions $1120 \times 1120 \times 770$ is used. After filling, the container can be removed and replaced with a new one.

Then, the layout diagram of a mobile wheeled platform representing a tractor cart with an automated system installed on it for harvesting apple crops was developed. The automated system consists of an industrial robot, a bagging system, a vision system, a control system, a vacuum generator, a power generator and a container. The problem of autonomous power supply of electrical equipment was solved by connecting the power generator to the power take-off shaft of the tractor.

Since the developed robotic complex uses only technical devices that have proven themselves in industrial operation, the use of this complex for harvesting apple crops seems promising and highly reliable.

The developed robotic complex for apple crop harvesting is an automated system composed of elements presented in the market of off-the-shelf industrial automation products. Production of such complexity can be organized in conditions with minimal requirements for its provision. Essentially, it requires a small assembly area with a point of connection to the power supply.

Maintenance and repair of the developed robot-technical complex also do not require special conditions. All necessary work can be performed directly in the field by replacing the failed element. Complex repairs of expensive elements, such as an industrial robot, are performed by service companies.

The expected practical efficiency of the proposed robotic complex is based on the implementation of the concept of an automated system, in which the autonomous work of the industrial robot and the system of packing harvested fruits is combined with the work of the human-controlled transport system, which is a mobile platform moving in coupling with a tractor. Because the human operator retains the functions of selecting the routes and driving modes of the transport system, it is expected to obtain a tangible productivity advantage over mobile robotic complexes that autonomously solve the tasks of navigation, route planning and driving modes.

Programming of the automated system is performed during equipment setup by setting the values of the relevant parameters and implementing calibration and machine-learning procedures.

It should be noted that this automated system does not use a neural network to solve the problem of object recognition. The stereo-image-based in-house software performs recognition of flat images of objects based on the image-processing algorithms built into the sensor. The sensor controller performs all the necessary calculations for positioning the gripper and planning the trajectories of the robotic arm.

This technical solution to the problem of object recognition is possible for the harvesting of the fruit of columnar apple crops. It is the convenient location of fruits along the trunk and the absence of branches with foliage cover that allows to reliably solve the problem of fruit recognition and perform operations for their automated harvesting.

Further development of the proposed robotic complex for harvesting apple crops can be aimed at increasing the level of automation of work related to the packaging of finished products. For this purpose, additional devices equipped with sensors and controlled electric drives can be used.

**Author Contributions:** Conceptualization, O.K., S.G., E.B., S.K., I.B. and K.L.; methodology, O.K.; validation, S.G., E.B. and I.B.; resources, S.K.; writing—original draft preparation, O.K.; writing—review and editing, E.B. All authors have read and agreed to the published version of the manuscript.

**Funding:** This research received no external funding.

**Institutional Review Board Statement:** The study was conducted in accordance with the Declaration of Helsinki, and the protocol was approved by the Ethics Committee of 20200901a.

**Informed Consent Statement:** Informed consent was obtained from all subjects involved in the study.

**Conflicts of Interest:** The authors declare no conflict of interest.

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
