# Peer review of "Robotic Complex for Harvesting Apple Crops"

_robotics, doi:10.3390/robotics11040077_

Round 1
Reviewer 1 Report
This paper article deals with the concept of building an automated system (robotic complex) for harvesting apple crops. Several of its components are described.
The motivation, that is proposing a solution to the problem of labor shortage during the harvest season in agriculture, is compelling. However there are remarks as explained next.
Major remarks
1. The authors are encouraged to present some data regarding the magnitude of the demand for apples in order to better justify the need to automate apples’ harvesting.
2. In the Introduction section 1, in a new paragraph, the authors are expected to clearly summarize the novelties of this work regarding the robotic complex they propose.
3. The authors, in line 115, speak of autonomy of their proposed robotic complex but they do not describe in the manuscript how they have achieved the aforementioned autonomy or, at least, how they can pursue it. The authors are expected to make some comments regarding the autonomy of their proposed robotic complex.
4. A large part of the manuscript presents methods in a descriptive manner rather than in a quantified manner. For instance, in line 144 it reads: “The vacuum grippers are very fast …” but the authors do not quantify numerically how fast the vacuum grippers are. Likewise, in line 208-209 it reads: “However, the best results in terms of time are shown by the vacuum device ...”; however, the authors do not quantify how good the best results are. The same problem (i.e. descriptive rather than quantified presentation) appears over and over again in the manuscript. The authors are expected to make some comments regarding any quantification of their proposed robotic complex.
5. In seems that the proposed robotic complex is not tested “on the field”. Hence, there is no evidence of its effectiveness, comparatively. Even the Abstract reads: “The developed automated system will have a high degree of reliability ...”. The authors are expected to make clear comments regarding the practical effectiveness of their proposed robotic complex.
6. It appears that the authors have not developed any of the components they use to develop their proposed robotic complex. Rather, it seems that they have used off-the-self components to develop it. The authors are expected to make comments regarding the development of their proposed robotic complex.
Minor remarks
1. The second sentence in the Introduction section, in line 26, reads: “This problem gives rise to other related problems associated with labor migration”. This sentence does not sound very relevant, therefore it should be either rephrased or omitted altogether.
2. In lines 39-40, it reads: “To solve this problem, the scientific literature distinguishes three main scientific directions. These include modeling, computer vision, machine learning, and robotics”. That is, four scientific directions are mentioned instead of three scientific directions. The authors are expected to correct, accordingly.
3. Perhaps, a relevant reference the authors may include in the Introduction section is the: "An overview of cooperative robotics in agriculture”, Agronomy 2021.
4. In line 131 the authors state that: “The lifting capacity of the robot is 30 kilograms”. Such a capacity is far too large to lift a single apple. The authors are expected to make some comments regarding the lifting capacity of their robot.
5. There are many examples of bad use of the English language. For instance, in line 79 it reads: “In the paper [35] authors …”; the authors should probably consider writing: “In [35] the authors …”. Furthermore, in line 81 it reads: “In [33] presents …”; the authors should probably consider writing: “The work in [33] presents …”. Furthermore, in line 88-90 it reads: “The Financial University under the Government of the Russian Federation (Moscow) with other agrarian and technical universities to conduct research into the automation of fruit and vegetable harvesting [8, 23, 26, 31]”; it seems that a verb is missing in the aforementioned sentence, hence the authors are expected to fill in the correct verb. Furthermore, in lines 129-130 it reads: “The task of the robot is to bring the gripper to the fruit and tear the apple off the tree …”; the authors should rewrite the aforementioned sentence in correct English. In lines 142-143 it reads: “The soft surface of the vacuum suction pad allows not to damage the apple…”; perhaps, the authors should consider writing: “The soft surface of the vacuum suction pad avoids damaging the apple…”. In lines 172-173 it reads: “In this case time saving in the process automation can be obtained both by ...”; perhaps, the authors want to write: “In this case, time saving in the process automation can be achieved both by ...”, that is replace the word “obtained” with the word “achieved”. In line 180 it reads “If you want to fully simulate human work, …”; the authors better rephrase the aforementioned sentence, impersonally. The authors are expected to read carefully the entire manuscript and improve the use of the English language everywhere.
6. There are some typographic errors. For example, in line 102 it reads “branch‐ ing”; the authors would probably wanted to write: “branching”. The authors are expected to read carefully the entire manuscript and correct all typographic errors.
Author Response
Dear reviewer!
Thank you for your constructive and kind comments on this article.
We are confident that working on these comments will help to improve this article.

Reviewer 2 Report
- Line 41 and 45 and other similar instances: Authors should cite the previous works properly. E.g. Author Surname et al. [32] mentions that the robot programming is a......
- Conclusions based on the existing works are not properly presented. It should be clearly brought up as to what are the limitations of the existing works and what part of that is being attempted in the present work.
- Fig 2 and 3 should be re-drawn using some standard CAD package. At present it is not depicting the developed system properly.
- Fig 6 and 7 should use markers to explain various elements of the system
- Line 159-164: The robot receives information about the position of apples and their coordinates from the vision system... How the vision system works is not presented or discussed in the work. What are the possible orientations? What are the preferred orientations?
- Line 165: Applied scientific results published by the authors earlier will be taken into account in the developed automated system [2, 16, 18‐23, 27]...it does not give any idea as to how the problem solving will be attempted.
- Line 220: The force of detachment of the stalk from the branch is about 2 kgf, this force is greater than the holding force of the apple, therefore we focus on this figure. Taking into account the safety factor of 1.5, the vacuum gripper has to provide a force of not less than 3.5 kgf. A quick calculation with this factor of safety gives results as 3 kgf. Please check and correct/ explain.
- Page 7-8: Rather than explaining too much about the parts/ components manufacturer's specifications, focus more on how they are used in the present work. Moreover, this description should not be a part of the results section. Authors should make a separate section for methodology.
- Results section does not present any case studies. Authors should include at least 1-2 such studies to showcase how the proposed system works.
- The Discussion section should be merged with the Results section.
- The Conclusions section should be improved by adding relevant descriptions of the results.
- Grammatical errors are present throughout the paper. E.g.
When (While) picking apples(,) an important task is to.... The packaging system is a system of transporting apples through an flexible... The article must be read thoroughly and all such errors should be corrected.
Author Response

(The authors gave the same response as above.)

Reviewer 3 Report
Call outs of references are too brief. For example, on line 28, "[5] authors" should be "Bu et al. [5]". Similarly on lines 41,45,49,51,56,58,60,61,63,64,68,69,71,75,77, 79, and 81.
The text that refers to Figure 18 should be improved. For example, starting in line 330, "Figure 18 shows the main elements or the robot complex, which includes the SmartPAD used to program robot movement." The sentence in lines 336 and 337 is ambiguous and should be removed.
The order of sections 4 and 5 should be exchanged. Conclusion is usually the final section of a manuscript.
Author Response

(The authors gave the same response as above.)

Round 2
Reviewer 1 Report
1. In my (major) remark “The authors, in line 115, speak of autonomy of their proposed robotic complex but they do not describe in the manuscript how they achieve the aforementioned autonomy or, at least, how to pursue it” the authors have responded writing the text “On the issue of ensuring the autonomy of our proposed robotic complex, there is information in the first paragraph of section 2. Methods: The problem of autonomous power supply of electrical equipment is solved by connecting the electric generator to the power take-off shaft of the tractor”. The previous text: a) is not located in the first paragraph of section 2, and b) it does not answer my remark; the latter regards mainly “autonomy” of decision-making /planning rather than power autonomy. Perhaps my remark was not clear. The authors are expected to make some relevant comments.
2. In my (minor) remark “In lines 39-40, it reads: “To solve this problem, the scientific literature distinguishes three main scientific directions. These include modeling, computer vision, machine learning, and robotics”. That is, four scientific directions are mentioned instead of three scientific directions. The authors are expected to correct, accordingly” the authors have not responded. The authors are expected to respond.
Author Response

(The authors gave the same response as above.)

Reviewer 2 Report
Revised manuscript is fine.
Author Response

(The authors gave the same response as above.)
